# ZnO Films Incorporation Study on Macroporous Silicon Structure

**DOI:** 10.3390/ma14133697

**Published:** 2021-07-01

**Authors:** Lizeth Martínez, Godofredo García-Salgado, Francisco Morales-Morales, Bernardo Campillo, Angélica G. Hernández, Tangirala V. K. Karthik, María R. Jiménez-Vivanco, José Campos-Álvarez

**Affiliations:** 1Tepeji Graduate School, Industrial Engineering, Autonomous Hidalgo State University, Av. del Maestro No. 41, Col. Noxtongo 2ª Sección, Tepeji del Rio, Hidalgo 42855, Mexico; angelica_hernandez@uaeh.edu.mx (A.G.H.); enkata_tangirala@uaeh.edu.mx (T.V.K.K.); 2Semiconductor Devices Research Center, Autonomus University of Puebla, Ciudad Universitaria, Puebla Pue 72570, Mexico; godgarcia@yahoo.com (G.G.-S.); kasslaa@hotmail.com (M.R.J.-V.); 3Optical Research Center, A. C., Loma del Bosque 115, Col. Lomas del Campestre, León 37150, Mexico; fcomm9@gmail.com; 4Faculty of Chemistry, Metallurgical Engineering Department, Autonomous National University of Mexico, Mexico City 04510, Mexico; bci@icf.unam.mx; 5Institute of Physical Sciences, Autonomus National University of Mexico, Cuernavaca, Morelos 62210, Mexico; 6Institute for Renewable Energy, Autonomus National University of Mexico, Priv. Xochicalco S/N, Temixco, Morelos 62580, Mexico; jca@ier.unam.mx

**Keywords:** zinc oxide, macroporous silicon, ultrasonic spray pyrolysis, spin coating, sol–gel

## Abstract

In the present work, we developed hybrid nanostructures based on ZnO films deposited on macroporous silicon substrates using the sol–gel spin coating and ultrasonic spray pyrolysis (USP) techniques. The changes in the growth of ZnO films on macroporous silicon were studied using a UV-visible spectrometer, an X-ray diffractometer (XRD), scanning electron microscopy (SEM) and atomic force microscopy (AFM). XRD analysis revealed the beneficial influence of macroporous silicon on the structural properties of ZnO films. SEM micrographs showed the growth and coverage of ZnO granular and flake-like crystals inside the pores of the substrate. The root mean square roughness (RMS) measured by AFM in the ZnO grown on the macroporous silicon substrate was up to one order of magnitude higher than reference samples. These results prove that the methods used in this work are effective to cover porous and obtain nano-morphologies of ZnO. These morphologies could be useful for making highly sensitive gas sensors.

## 1. Introduction

At present, the effort to obtain nanostructures/hybrids from semiconducting materials has attracted considerable interest in the scientific community due to the possible superior functional properties that they could present compared to the individual components assembling composites. Among the related experimental works, zinc oxide (ZnO) stands out due to its morphological and physical properties. ZnO is a semiconductor material with a direct bandgap of 3.37 eV at room temperature and a high excitation binding energy of 60 meV [1]. On the other hand, ZnO has been recognized as a promising material for some electronic and optoelectronic applications, such as solar cells [2], transparent conductors [3,4], piezoelectric transducers [5], memristors [6], surface acoustic wave devices [7] and gas sensors [8]. In addition, ZnO offers distinct advantages over other metal oxides due its abundance, non-toxic nature and easy synthesis. There are different methods for ZnO synthesis, among which are pulsed laser deposition (PLD) [9], chemical vapor deposition (CVD) [10], the microwave [11] and hydrothermal methods [12], RF magnetron sputtering [13], the sol–gel process [14], ultrasonic spray pyrolysis (USP) [15,16] and homogeneous precipitation [17]. ZnO deposited on porous samples can present diverse morphologies such as nanorods [18], nanobelts [19] and nanowires [20]. The advantages of using such morphologies derive from the large surface-to-volume ratio, high specific area and surface roughness that they can present [21]. Furthermore, these ZnO morphologies have been used to realize sensors for various gases. These include CO_2_, NO_2_, CO, NH_3_, O_3_, H_2_S, C_2_H_5_OH and H_2_ [17,21,22,23,24,25,26,27].

Among the porous nanostructures, porous silicon (PS) has a rough surface morphology which provides good sites for reducing the stress of ZnO growth [17]. In particular, PS has a structure with tunable pore dimensions and good compatibility with IC technology [28]. Its open structure and large surface area make PS a good candidate for templates and gas sensing applications [29]. For gas sensing applications, it is convenient to describe the sensing process as a surface mechanism in which the interactions among the sensor and gas molecules are defined by its surface area and morphology [30,31]. 

Recently, the detection of gases using ordered porous metal oxide nanostructures has been reported, where they can considerably facilitate gas diffusion to increase the sensing response and kinetics [30,32]. On the other hand, the incorporation of ZnO on macroporous silicon (mPS) is possible for enhancing the sensitivity of hybrid structure (ZnO/mPS) photodetectors, and also enhancing and tuning their photoconductivity and optical properties [33]. Additionally, a non-enzymatic H_2_O_2_ sensor based on mPS for environmental and industrial applications has been reported, in which the surface area of the macroporous silicon structure plays an important role in the detection of H_2_O_2_ [34]. Apart from the gas sensing application, ZnO–mPS has been widely used for optoelectronic devices as well. Kumar et al. deposited Al-doped ZnO films on PS substrates by a RF sputtering process and demonstrated a broadening in the PL emission band, which led to white light emission [35]. ZnO and mPS offer high and promising properties in different fields of application.

In the effort to propose a new functional material for possible gas sensing applications, we have deposited ZnO films on mPS using the sol–gel spin coating and ultrasonic spray pyrolysis (USP) techniques. The effect of the ZnO films on the porous structure has been discussed using the optical, morphological and structural characterization techniques. The results prove that the methods used in this work are effective for covering porous and obtaining nano-morphologies of ZnO. 

## 2. Materials and Methods

### 2.1. Fabrication of Macroporous Silicon Substrates

The mPS samples were fabricated using p-type boron-doped crystalline silicon (cSi) wafers, in (100) orientation with a resistivity of 1–20 Ω·cm. The anodization process was carried out in 1.5 cm × 1.5 cm wafers, using an electrolyte based on hydrofluoric acid (Meyer, 48 wt.%) [HF] and dimethylformamide (Fermont, 99.9%) [DMF: HCON(CH_3_)_2_] in a volumetric ratio of 1:3. The porous structure was fabricated by applying a current density of J = 10 mA/cm^2^ for 10 min. The mPS samples were oxidized at 400 °C for 30 min to stabilize the porous silicon surface.

### 2.2. Deposition of ZnO Thin Films by the Sol–Gel Spin Coating Method

The deposition of ZnO films by sol–gel spin coating method consisted of two stages: (1) synthesis of ZnO precursor solution by sol–gel method, and (2) deposition of the ZnO precursor solution using the spin coating technique. In the first stage, 0.2 M of dihydrated zinc acetate (98.0–100%, Alfa Aesar) [Zn(CH_3_COO)_2_·2H_2_O] was dissolved in 50 mL ethanol (99.7%, Fermont) [C_2_H_5_OH], and kept under magnetic stirring for 1 h. Later, monoethanolamine (99%, Meyer) [MEA: HOCH_2_CH_2_NH_2_] was added in a 1:1 molar ratio during stirring and kept there for 1 h, until a clear solution was obtained. Finally, dihydrated MEA/zinc acetate was kept in an ultrasonic bath (Mikel’s, TLU-3) at 50 °C for an hour and subsequently left at room temperature for 48 h for hydrolysis [35]. 

In the second stage, mPS substrates were immersed in the previously prepared solution and kept in an ultrasonic bath (Mikel’s, TLU-3) for 30 s. Subsequently, ZnO excess was removed and distributed on the mPS surface using the spin coating method (Laurell, WS-650-23) at 3000 rpm for 30 s. In order to remove the organic solvents of the samples, the substrates were heated and dried on a hot plate at 100 °C for 5 min. The above process was repeated seven times to cover the mPS. Finally, the samples were subjected to heat treatment (ARSA, AR-340) at 400 °C for 30 min.

### 2.3. Deposition of ZnO Thin Films by the USP Method

This deposition method consisted in the preparation of the ZnO precursor solution, in which 0.2 M of zinc acetate dihydrate (≥98%, Alfa Aesar) [Zn(CH_3_COO)_2_·2H_2_O] was diluted in 50 mL ethanol (99.7%, Fermont) [C_2_H_5_OH] and kept under magnetic stirring for 1 h at room temperature. During deposition the substrate temperature was kept at 260 °C ± 3 °C for 10 min. The pressure and carrier gas flow (N_2_) were maintained at 276 kPa and 0.09 L/min, respectively [36]. Finally, the samples were subjected to heat treatment (ARSA, AR-340) at 400 °C for 30 min. To obtain the control samples, the deposition processes were carried out on glass substrates of 1.5 cm × 1.5 cm. Table 1 shows the names of the prepared samples.

### 2.4. Characterization of ZnO Thin Films

The optical transmittance of the ZnO films deposited on glass substrates was analyzed using a UV-Vis spectrometer (VE-5100UV, VELAB, CDMX, México). The structural properties of the ZnO films deposited on glass and mPS substrates were investigated by an X-ray diffractometer (XRD) (AXS D8 Discover, Bruker, Karlsruhe, Germany) employing CuKα radiation and λ = 1.54 Å. The morphological surface and topography were analyzed with scanning electron microscopy (SEM) (JEOL JSM 7401F, Hitachi High-Tech Canada. Inc., Toronto, Ontario, Canada) and an atomic force microscope (AFM) (Park NX10, Park Systems Inc., Suwon, Corea). AFM measurements were conducted at room temperature in non-contact mode. The cantilever was made out of silicon with a spring constant of 42 N/m and a nominal tip apex radius of 10 nm. Measurements were performed for a scan size of 10 × 10 µm^2^ with a resolution of 256 × 256 pixels. The analysis of AFM measurements was analyzed with the help of the XEI program.

## 3. Results and Discussion

In order to check the quality of the ZnO films, the transmittance was measured in the range 350–750 nm. Figure 1 shows the optical transmittance, bandgap and cross-sectional SEM images of ZnO films deposited on glass substrates (Z1G) by the sol–gel spin coating and USP (Z2G) techniques. The transmittance spectrum of the ZnO films (Figure 1a) showed an optical average transparency over 84% in the visible range and an accentuation in the region of 360 nm (Z1G) and 370 nm (Z2G) corresponding to sol–gel spin coating and USP deposition, respectively. The sharp ultraviolet absorption edges of the samples at a wavelength close to 360–370 nm corresponds to the intrinsic bandgap energy of ZnO [35,37]. The decrease in the average transparency can be attributed to irregularities in the grain boundaries of ZnO and scattering due to surface roughness [38]. About 80% transparency in ZnO films has been reported to be sufficient for optoelectronic applications [39]. Furthermore, Mirela Suchea et al. studied ZnO transparent films deposited on cSi and glass substrates and found that the gas sensing characteristics of the ZnO films were highly influenced by surface morphology [40].

From the experimental transmittance spectra of ZnO films, which have an average thickness (Figure 1b) of 176 nm (Z1G) and 412 nm (Z2G), the absorption coefficient (α) and its relation to the bandgap (Eg) were determined using the equation αhv = A (hv-Eg)^1/2^, where A is the constant, Eg is the optical bandgap, v is the incident radiation frequency and finally h is Planck´s constant [41,42]. Inside Figure 1a, we can see the Tauc plot (αhv)^2^ vs. photon energy (hv). The Eg value of ZnO film was obtained by extrapolating the straight line portion of plot to zero absorption coefficient. The bandgap of ZnO film was found to be Eg = 3.30 eV for sol–gel spin coating (Z1G) and Eg = 3.23 eV for USP (Z2G) technique, respectively. The Eg values obtained are similar to bulk ZnO, which reveals the good quality and impurity-free nature of ZnO films prepared by sol–gel spin coating and USP [43,44].

The crystallinity of ZnO films on mPS was investigated using the XRD technique. Figure 2 shows the XRD patterns of ZnO films deposited on glass and mPS substrate by the sol–gel spin coating (Z1G, Z1P) and USP methods (Z2G, Z2P), respectively.

The XRD spectra showed the presence of (100), (002), (101) and (110) peaks, demonstrating that the samples Z1P and Z2P were polycrystalline (Joint Committee for Powder Diffraction Studies (JDPS) No. 36-1451) [10]. The diffractograms also show the presence of a (200) peak in the ZnO films deposited on mPS (Z1P, Z2P), indicating the presence of silicon (Si) in the substrate [45]. It can be seen that the most samples have a second maximum diffraction peak along the (002) plane. The above demonstrates that the ZnO films crystallized in the wurtzite phase of the hexagonal structure with a preferential orientation towards the c-axis, which is perpendicular to the substrate surface [46]. The c-axis orientation corresponds to the densest plane and means that the ZnO films have piezoelectric properties [47]. Gonzalo Alonso Velázquez-Nevárez et al. corroborated that the ZnO films with orientation (002) grown at 400 °C were the most appropriate to achieve lowest resistivity (7.1 Ω·cm). They suggested that appropriate annealing temperature to achieve smallest resistivity was 400 °C, which may have been related to the high amount of oxygen vacancies [48]. In addition, F. Fitriana et al. found that (002)-oriented ZnO has high potential for a highly sensitive CO and NO sensor [49].

The average crystalline size (D) and the strain-induced broadening (ε) of ZnO films were calculated from X-ray diffraction using the Scherrer equation and the Williamson–Hall method (W–H) [50,51]. According to Williamson and Hall, diffraction broadening is due to crystallite and strain contribution [52]. The strain-induced broadening of ZnO films was calculated using the formula ε = β/4tanθ; where β is the full-width half-maximum (FWHM) [51]. The values are shown in Table 2. The crystallite sizes for Z1G and Z1P were 20 nm and 15 nm, respectively. The decrease in the crystal size from Z1G to Z1P indicates an improvement crystallization of ZnO films when they are deposited on mPS. We can observe that the opposite occurs for Z2G and Z2P samples. The crystallite size from Z2G to Z2P was 6 nm to 8 nm, respectively. The increase in crystal size could be due to the deposition process not allowing time to connect small nanocrystals on the porous substrates. These results have been corroborated by the reports of Hong Cai et al. [53] and Tae-Bong Hur et al. [54]. They attributed the increase in crystal size to a decrease in strain and the skeleton of porous silicon, in which the steady-state regime relaxes stress [53,54].

The role of the substrate is very important to limit the direction and rate growth of ZnO films [18]. However, the crystallite size could be also influenced by various factors such as impurities, defects, temperature and synthesis environment conditions [55,56]. That may be the reason why the values data are scattered and there is present a trend of decreasing crystallite size when the FWHM increases.

The SEM images were obtained in order to study the porous substrate effect and the deposition method’s influence on ZnO films morphology. Figure 3 shows the top (left side) and cross sectional (right side) micrographs of the samples. The bare mPS substrate is shown in Figure 3a,b. These micrographs reveal the mPS substrate morphology displaying an irregular distribution of round pores of ~1 µm diameter (Figure 3a). The cross-sectional image of the mPS substrate shown in Figure 3b corroborates the pore diameter and the mPS film thickness at ~2 µm.

In Figure 3c,d, we can see the ZnO films deposited on the mPS substrate by the sol–gel spin coating method (Z1P). The top-sectional image shown in Figure 3c,d reveals the completely decorated pore presenting the formation of granular ZnO films on the entire surface of the mPS substrate. In addition, the fracture of some pores can be observed, which surely could have been caused by sonication during the ZnO deposition process. In Figure 3e,f, we have observed the ZnO deposited on mPS substrate by USP method deposition (Z2P). We found that ZnO films exhibit a dense structure covering the mPS surface. Figure 4 shows the SEM magnification of ZnO deposited on mPS substrates (Z1P, Z2P). 

Figure 4a shows the infiltration of ZnO films inside the pore; in this way, we can confirm the ZnO’s granular shape and the connectivity of nanocrystals covering the pore. The crystal connectivity enhances the bonds existing in the ZnO films, which are favorable for adsorption of gas molecular [57]. Figure 4b shows the ZnO films deposited on mPS structure by the USP method. We found that small flake-like nanocrystals grew on the macroporous silicon skeleton in vertical form, so that the XRD (002) diffraction peak increased somewhat (Figure 2b). These flake-like crystals could have been formed due to the porous silicon substrates providing nucleation centers for initial formation and the free energy at nanometric scale inducing the ZnO crystal growth [13,58,59,60]. SEM micrographs gave us proof that the methods used in this work are effective for covering porous and obtaining nano-morphologies of ZnO. 

Figure 5 shows the two dimensional AFM images taken at a scan area of 10 × 10 µm^2^ (a) and the comparison of depth profiles (b) of ZnO deposited via sol–gel spin coating (Z1G, Z1P) and USP (Z2G, Z2P) techniques. Figure 5b shows the longitudinal increase in waviness in the ZnO–mPS surface profile is higher than ZnO–glass samples. The roughness mean square (RMS) values from the ZnO films deposited on glass substrates were 16 nm (Z1G) and 10 nm (Z2G). The above results are similar to other reports related to surface roughness studies by AFM [61,62]. RMS values from ZnO films deposited on mPS were 244 nm (Z1P) and 194 nm (Z2P), respectively. Such results show that ZnO deposited on mPS increased the roughness surface up to one order of magnitude compared to the ZnO deposited on glass substrates. These results strengthen earlier studies by U. Salazar-Kuri et al. and confirm the increase in surface roughness of ZnO deposited on mPS [63]. From the SEM and AFM results above, it can be seen that the mPS substrate can be used as a good template to increase the surface roughness of ZnO films; in addition, its porous structure can improve the growth of ZnO.

## 4. Conclusions

In this paper, it was possible to deposit ZnO films on mPS substrates by the sol–gel spin coating and USP methods. The ZnO pore decoration of mPS substrates revealed the formation of granular nanocrystals for the sol–gel spin-coating method and flake-like crystals for the USP method. The crystal size and shape were described in terms of strain-induced broadening and the influence of mPS. The roughness surface of the ZnO films on mPS substrates increased up to one order of magnitude compared to the ZnO deposited on glass substrates. SEM and AFM analysis confirmed that the mPS substrate can function as a good template for increasing the ZnO surface roughness and as a template for ZnO crystal growth. This study can lead the way to more extensive studies of the employment of ZnO with a mPS structure in the sensors field, including gas sensors and biosensors as well as the sensing of different substances as ethanol, drugs and explosives.

## Figures and Tables

**Figure 1 materials-14-03697-f001:**
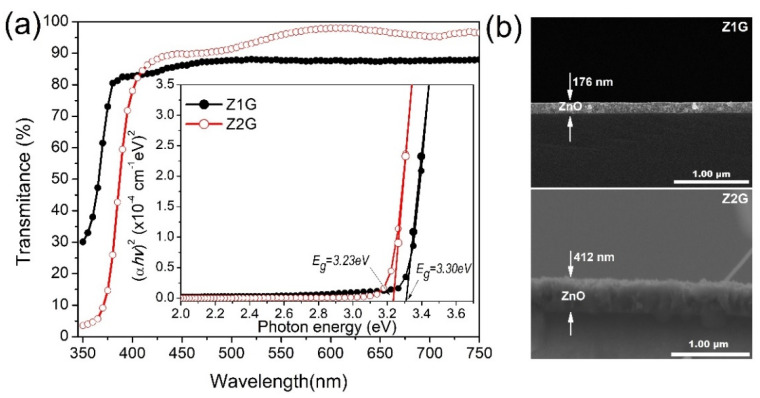
(**a**) Optical transmittance and band gap properties; and (**b**) cross-sectional SEM images of ZnO films deposited on corning glass by sol–gel spin coating (Z1G) and USP (Z2P) methods.

**Figure 2 materials-14-03697-f002:**
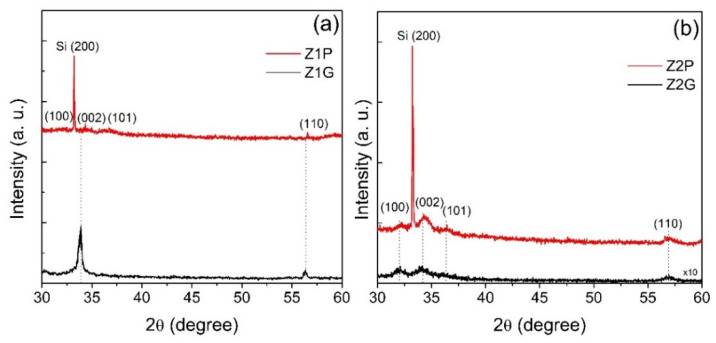
XRD patterns of ZnO films deposited by the sol–gel spin coating (**a**) and USP (**b**) methods on glass (Z1G, Z2G) and macroporous silicon (Z1P, Z2P) substrate.

**Figure 3 materials-14-03697-f003:**
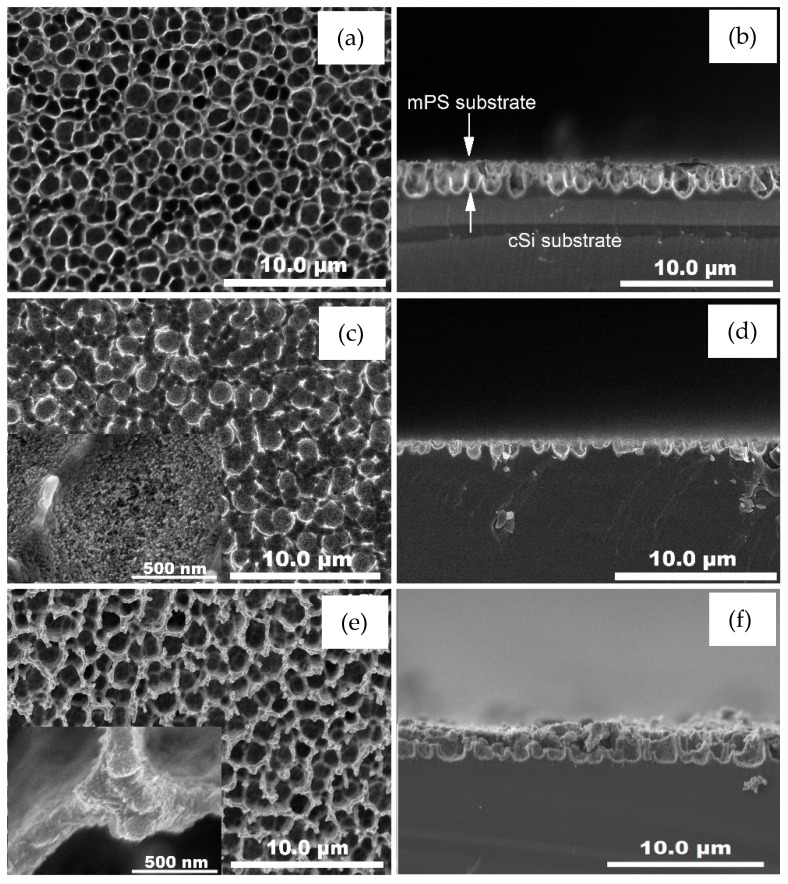
SEM micrographs of the top (left side) and cross (right side) sectional of bare mPS substrate (**a**,**b**); ZnO films deposited on mPS by sol–gel spin coating Z1P (**c**,**d**), and USP, Z2P (**e**,**f**) methods.

**Figure 4 materials-14-03697-f004:**
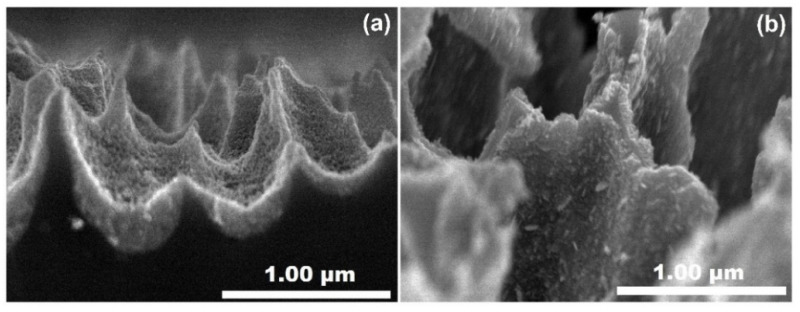
SEM micrographs of cross-sectional images with higher magnifications corresponding to Z1P (**a**) and Z2P (**b**).

**Figure 5 materials-14-03697-f005:**
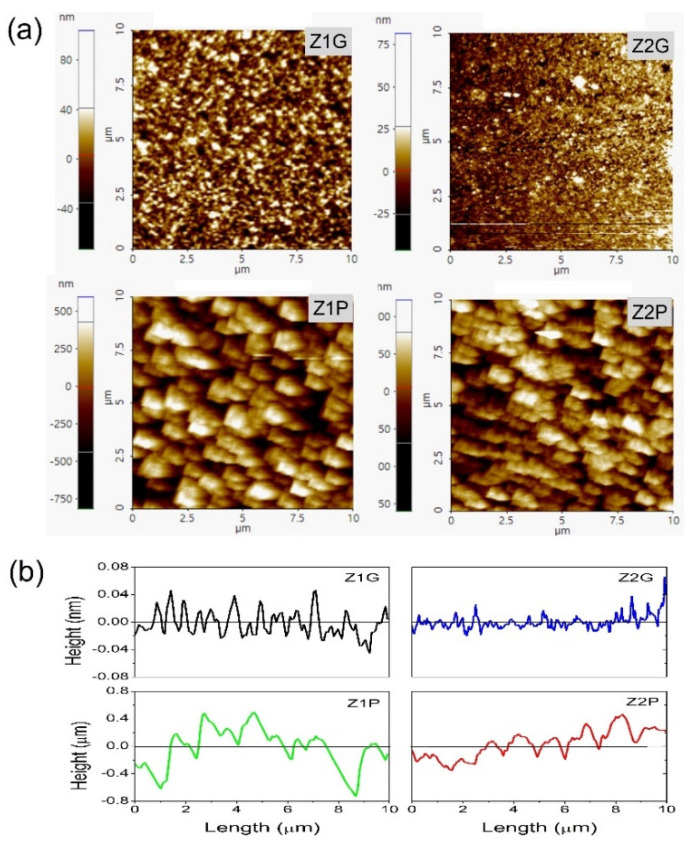
Atomic force microscopy images (**a**) and height profile (**b**) of ZnO deposited by sol–gel spin coating (Z1G, Z1P) and USP (Z2G, Z2P) techniques.

**Table 1 materials-14-03697-t001:** Summary of samples fabricated.

Sample	Characteristics
Z1G	ZnO on glass substrate by sol–gel spin coating method
Z1P	ZnO on mPS substrate by sol–gel spin coating method
Z2G	ZnO on glass substrate by USP method
Z2P	ZnO on mPS substrate by USP method

**Table 2 materials-14-03697-t002:** Summary of structural parameters of ZnO thin films deposited on glass and mPS substrates by sol–gel spin coating and USP methods.

Sample	FWHM (002)	2 θ (°)	Crystallite Size (nm)	ε
Z1G	0.4213	33.8145	20	0.100826
Z1P	0.5312	34.4291	15	0.126851
Z2G	1.3099	34.0601	6	0.313116
Z2P	0.9811	34.4287	8	0.234288

## Data Availability

Not applicable.

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
