# Peer review of "ZnO Films Incorporation Study on Macroporous Silicon Structure"

_materials, 2021, doi:10.3390/ma14133697_

Round 1

Reviewer 1 Report

Martinez L et al proposed ZnO-Silicon hybrid porous micostructure having high surface area for sensor applications. This work is interesting and could benefit its field. However, before I can recommend the publication of this article, the authors should address the problem below;

  1. There are typing errors and incorrect phrases throughout the manuscript. For example, line 19: “..we have development hybrid nanostructures based on ZnO films..”, it should be either “..we have developed..” or “ ..we report the development of..”. Line 29: “These morphologies can offer the possibility to fabricate highly sensitive gas sensors.” It can be polished as “These morphologies could be useful for making highly sensitive gas sensors.”  I suggest the english of the manuscript can be polished further.
  2. The authors can add another example of interesting 2-dimensional structured ZnO-based for electronic application which is memory devices (DOI: 10.1109/TNANO.2020.3029588)
  3. Line 57, 66. “Surface roughness” is not exactly accurate. The authors should use “surface area” instead, since in gas sensing mechanism the reactions happens on the surface; hence, the sensitivity depends on the total effective area that could facilitate the reactions. The authors can study and cite this paper regarding surface area study (DOI:10.1007/s10853-020-04659-7)
  4. For consistency, the authors should provide SEM images with the same observation parameter to ensure accurate comparison. For example; Fig.1b should have the same magnification, and Figs. 3b, d, and f should have the same tilt angle.
  5. SEM and AFM are qualitative analysis, which they are not sufficient to determine which microstructure is better. I suggest the authors conduct BET surface area analysis to confirm which sample (Z1P or Z2P) that actually has better morphology.
  6. The thickness of the samples are not the same (nor similar); hence, comparing these samples could be misleading. The thickness can significantly alter almost every properties of the ZnO (microstructure, band gap, crystal orientations, etc.). Samples made with the same substrate can be compared if they have the same thickness (Z1G and Z2G). I suggest the authors should do deposition rate test for either one of the methods (sol-gel or USP) then re-fabricate the samples to have similar thickness to the control sample. For example, if the authors choose to pick 176 nm Z1G as the control sample, then Z2G should be deposited about 176 nm as well; consequently, Z2P can be prepared using the rate of whichever Z2G used -or vice versa.

Henceforth, at the present form, the analysis reported in the manuscript cannot support the given conclusion. 

Reviewer 2 Report

Dear Authors, in your interesting manuscript, the following points should be added/changed to further improve it:

  1. Introduction: I have a suggestion to the sentence [44-48]. Please add an example of microwave synthesis as an example of a method to obtain zinc oxide (e.g. extensive review article DOI:10.3390/nano10061086)
  2. Introduction: Please add some examples of gases that can be detected using sensors based on zinc oxide.
  3. Materials and Methods: There is no information about the reactants (purity, producer). Whether the reagents were further purified?
  4. Materials and Methods - Deposition of ZnO thin films by sol-gel spin coating method: Which zinc acetate was used, dehydrated [83, 98] or dihydrated [84,98] ?
  5. Materials and Methods - Deposition of ZnO thin films by sol-gel spin coating method: Please provide information about stirring (rpm, model, manufacturer) [85].
  6. Materials and Methods - Deposition of ZnO thin films by sol-gel spin coating method: Actors must report the exact volumes/amounts of reagents used. Please complete the description.
  7. Materials and Methods - Deposition of ZnO thin films by sol-gel spin coating method: Whether the solution was stirred during the addition of monoethanolamine?
  8. Materials and Methods - Deposition of ZnO thin films by sol-gel spin coating method: Please add information about the size of the mPS substrates.
  9. Materials and Methods - Deposition of ZnO thin films by sol-gel spin coating method: There is no information about the ultrasonic bath (power, model, manufacturer).
  10. Materials and Methods - Deposition of ZnO thin films by sol-gel spin coating method: There is no information about the laboratory furnace used to heat treatment the samples (model, manufacturer).
  11. Materials and Methods - Deposition of ZnO thin films by USP method: Please add information about the size of the glass substrates [103].
  12. Results: I understand that the authors have chosen to combine the results with the discussion ? [112]
  13. Results: I suggest to add SEM images with identical magnification (Figure 2b). Please consider adding a result of the layer thickness in the SEM images. This will make it easier for the readers to assimilate the results.
  14. Results: Please clarify meaning of the words “highly crystallinity” [154-156]? What is the proof of that?

Reviewer 3 Report

Martínez et al. present the development of hybrid nanostructures based on ZnO films deposited on macroporous silicon substrates using two techniques: sol-gel spin coating and ultrasonic spray pyrolysis (USP). The authors observed that the ZnO pore decoration of microporous silicon substrates leads to the formation of granular nanocrystals for the sol-gel spin-coating method, while for the USP method it was observed the formation of flake-like crystals. I recommend major revisions before a possible publication in Materials, as observed in the following comments:

  • The authors should pay attention to the subscripts and superscripts in the materials and methods section.
  • Although the authors just intend to present the fabrication methods and the characterization of the ZnO films and not the gas sensors, since the title mention the gas sensing applications and the main motivation for this work seems to be the applications of these films for gas sensing applications, the authors should well explain along the manuscript why these films are promising for gas sensing applications. As alternative the title could be changed, still decreasing the interest of the paper. In the case of adding these suggestions, a comment about the gas sensing application should also be included in the conclusions.
  • The quality of the images should be improved, and the size of the Figure 5 should be increased.

Reviewer 4 Report

The paper describes the structure, morphology and roughness of ZnO deposited on porous Si by two different processes (sol-gel and ultrasonic spray pyrolysis), using adequate characterisation methods. The results are clearly presented. However, there are some problems that authors are kindly requested to take into account:

  •  More details on the equipment and software of characterization equipment must be provided, in particular for AFM used to demonstrate the increase of roughness; 
  • It is a very large difference of the two values of FWHM provided. Did you calculated the mean crystallite sizes with these values? It is worthy to give these values to asses the differences in crystallization degree
  • What is the film from fig. 3b made of? It is the SiO2 film? 
  • There are no results or literature data to support possible gas sensing applications of ZnO coatings on porous Si. The increase of surface roughness of coatings on porous Si compared to glass is not enough to demonstrate the potential for gas sensing. It would be recommended to use BET measurements giving clear values of the porosity obtained by the two methods compared to porous Si wafer. 
  • Some corrections of English must be done; e.g.: ....functional properties that they could be presented...(in introduction), probably must be that they could present...Also in " The Figure 4a"  or " The Figure 5" it is correct to write Figure 4 or Figure 5, without " The". 

Round 2

Reviewer 1 Report

The authors have carefully addressed my comments. Moreover, they have revised the title and objective the study which is more suitable to the scope of the content. Hence, I can recommend the publication of this manuscript.

Reviewer 2 Report

The answers from authors and the revised manuscript is acceptable at present form.

Reviewer 3 Report

Accept in present form

Reviewer 4 Report

Dear Authors, thank you for the efforts to review the paper. You completions and information on the methodology as well as the additional bibliography clearly contributed to increase the quality of the paper and its originality.